# The Protection of Data Sharing for Privacy in Financial Vision

**Yi-Ren Wang [1],[†] and Yun-Cheng Tsai [2],[*],[†],[‡]**

1 Department of Data Science, Soochow University, No. 70, Linhsi Road, Shihlin District,
  Taipei City 111002, Taiwan; 09370008@gm.scu.edu.tw
2 Department of Technology Application and Human Resource Development,
  National Taiwan Normal University, 162, Section 1, Heping E. Rd., Taipei City 106209, Taiwan
* Correspondence: pecu@ntnu.edu.tw or pecu610@gmail.com
† These authors contributed equally to this work.

**Abstract:** The primary motivation is to address difficulties in data interpretation or a reduction in model accuracy. Although differential privacy can provide data privacy guarantees, it also creates problems. Thus, we need to consider the noise setting for differential privacy is currently inconclusive. This paper's main contribution is finding a balance between privacy and accuracy. The training data of deep learning models may contain private or sensitive corporate information. These may be dangerous to attacks, leading to privacy data leakage for data sharing. Many strategies are for privacy protection, and differential privacy is the most widely applied one. Google proposed a federated learning technology to solve the problem of data silos in 2016. The technology can share information without exchanging original data and has made significant progress in the medical field. However, there is still the risk of data leakage in federated learning; thus, many models are now used with differential privacy mechanisms to minimize the risk. The data in the financial field are similar to medical data, which contains a substantial amount of personal data. The leakage may cause uncontrollable consequences, making data exchange and sharing difficult. Let us suppose that differential privacy applies to the financial field. Financial institutions can provide customers with higher value and personalized services and automate credit scoring and risk management. Unfortunately, the economic area rarely applies differential privacy and attains no consensus on parameter settings. This study compares data security with non-private and differential privacy financial visual models. The paper finds a balance between privacy protection with model accuracy. The results show that when the privacy loss parameter $\epsilon$ is between 12.62 and 5.41, the privacy models can protect training data, and the accuracy does not decrease too much.

**Keywords:** financial vision; Gramian Angular Field (GAF); differential privacy; private aggregation of teacher ensembles (PATE); differentially private stochastic gradient descent (DP-SGD)

## 1. Introduction

Financial institutions are gradually moving towards a cooperative model of information sharing. The financial supervisory commission plans to provide financial API services. After obtaining customer authorization, banks can query customer or transaction information using financial APIs.

In addition to the financial API, in 2016, Google proposed a new data sharing technology —federated learning [1]. Joint members train the model individually. These members update the collaborative model by exchanging model gradients or parameters without original data. Federated learning can improve problems caused by data silos or data integration across enterprises. Federated learning has made significant progress in the medical field. For example, Taiwan recently participated in an international joint learning project. It successfully built a model with 94% accuracy to predict patients with COVID-O2 symptoms, leading to better clinical decisions [2].

Federated learning can be applied to finance. Data in the economic domain are similar to medical data. The value of data across agencies can be significantly enhanced by combining them. Financial institutions can provide customers with higher-value and personalized services, such as automated tracking and trading in stock or currency markets. AI models can also automate credit scoring and risk management and solve business challenges such as fraud [3,4]. However, most of these data contain a substantial amount of personal data. Leaks can have uncontrollable consequences, making data exchange and sharing difficult. With federated learning techniques, financial institutions can collaborate without exchanging private information [5].

Several companies such as IBM [6], Microsoft [7], or Amazon [8] have applied federated learning to provide customers with secure data analysis and AI model training to effectively mine potential value. For example, we now have some candlestick charts (Figure 1). Each of them represents a signal that may be bullish or bearish. These signals need to be labeled by experts, and then we can use these labeled data to build signal detection models. Therefore, data are sensitive information. However, the data labeled by each institution are limited. Data can maximize its benefits if information can be shared using federated learning techniques.

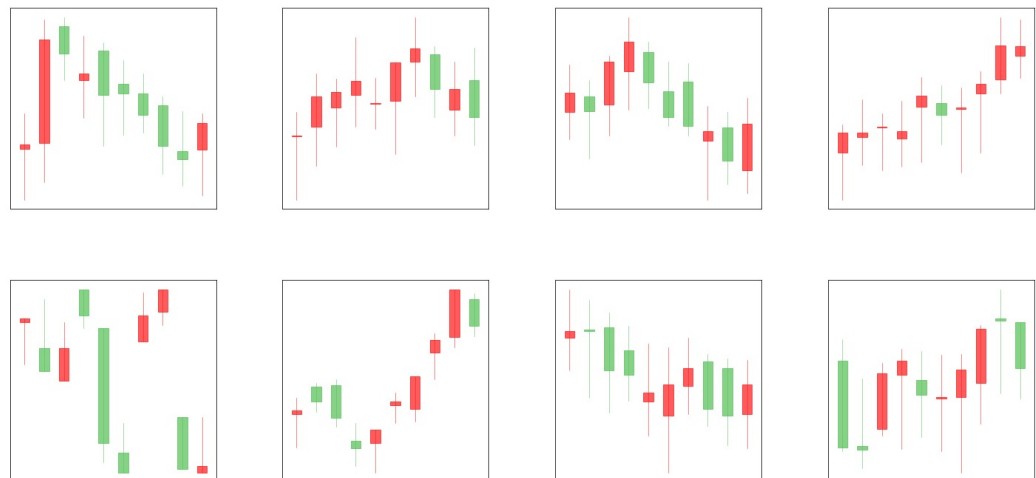

**Figure 1.** The candlestick charts. The red indicates a price increase; the green shows a price decrease.

Privacy protection is a crucial issue today, and many studies have considered the balance between big data and data privacy when conducting research. K-anonymity [9] is easy to understand but not sufficient for protecting private data. There are some success stories of unlocking anonymous data sets by using public data sets. For example, de-anonymized Netflix data were successfully studied by comparing Netflix's anonymized viewing history with IMDb data [10]. The anonymous Massachusetts Group Insurance Commission (GIC) database is another well-known case and includes the date of birth, gender, and zip code, all linked to voter registration records. The Massachusetts governor's record is fixed [11]. There are other similar publications. In addition to de-anonymization, M Fredrikson also identified training sets without public datasets. Nonetheless, the exposed model [12] shows no available training set. Release-only models also have privacy risks.

However, the parameters or gradients of the local model are exchanged for the federated learning model. There is also the risk of data leakage [13]. Therefore, additional privacy-preserving mechanisms are required. Many privacy-preserving methods can prevent the re-identification problem of a particular individual's private information caused by comparisons or queries with other databases. The most widely used one is differential privacy [14]. Many federated learning models are now used with differential privacy mechanisms to minimize the risk of data leakage [4,15]. More businesses are willing to cooperate in the context of personal data protection, creating more significant benefits. Differential privacy was proposed by Dwork et al. 2006 [16], reducing the probability of

data leakage by adding noise during model training. Training a model using a differential privacy mechanism makes it difficult to compare training data with public data, which is why it provides strong privacy protection. Currently, many well-known companies such as Apple, Google, and Microsoft use differential privacy to apply for data privacy [17].

Traditional financial trading usually relies on traders or experts to judge whether it is bullish or bearish by analyzing candlestick patterns and then deciding whether to buy or sell. With the development of deep learning technology, the topic of financial technology (FinTech) has become a focus of attention. The number of companies offering automated trading or tracking services is also growing. Before building a trade tracking model, it is necessary to consider the choice of indicators, such as the candlestick patterns mentioned above. Experts develop indicators such as sensitive data that can be leaked when trading models are released. The leakage of these sensitive data can cause immeasurable losses to businesses and customers. Because these data are valuable, someone hacks the model to obtain it. Therefore, protection mechanisms are essential.

Differential privacy is one of the most popular privacy-preserving mechanisms, but unfortunately, it is rare in finance. The advantage of differential privacy is that it provides strong privacy protection. However, differential privacy protects private data by adding noise and reducing model accuracy.

This study compares data security with non-private and personal models of financial vision. It also finds a balance between privacy protection and model accuracy. The results show that when the privacy loss parameter is between 12.62 and 13 5.41, the privacy models can protect training data, and the accuracy does not decrease too much.

## 2. Materials

### 2.1. Candlestick

Munehisa Homma designed the candlestick in the 17th century to visualize the price of rice [18]. It has become the most common chart in finance. A candlestick chart represents the highest price, the lowest price, the opening price, and the closing price (OHLC) in a specific period. Thus, it allows the investors to understand the market situation quickly and helps them make decisions. There are three parts of the candlestick chart introduced as follows, and Figure 2 is the structure of a candlestick:

1.  Real Body: Real Body represents the price difference between opening and closing.
2.  Shadow: There are two types of shadow. The upper shadow is the price difference between the highest price and the real body. The lower shadow is the price difference between the lowest price and the real body.
3.  Color: Color reveals the direction of market movements. The white or green body indicates a price increase; the black or red body shows a price decrease. However, the red body indicates a price increase, and the green body shows a price decrease in Taiwan.

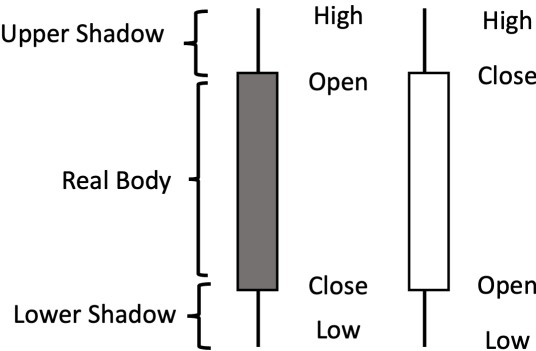

**Figure 2.** The structure of candlesticks charts.

A candlestick pattern refers to a group of images composed of several candlestick charts. Stephen W. Bigalow developed a simple way to interpret market signals [19]. For example, a Doji is a candle that forms when the opening and closing prices are the same or nearly the same. If the Doji appears at the top, it is overbought. In addition to the Doji, candlesticks have several basic combinations of patterns identified by traders with professional knowledge. The following section will introduce the GAF-CNN model, which is suitable for classifying images containing time-series information such as candlestick patterns.

### 2.2. GAF-CNN Model of Financial Vision

Wang and Oates first proposed GAF as a new framework for encoding time series into images [20]. GAF represents time-series data by replacing the typical Cartesian coordinates with the polar coordinate system and converting angles into symmetric matrices. The Gramian Angular Summation Field is a GAF that uses the cosine function. In the GASF matrix, each element is the cosine of the summation of the angles. GASFs can be transformed back to time-series data by diagonal elements, and compared to Cartesian coordinates, polar coordinates preserve absolute temporal relations.

CNN models are well-known algorithms that take advantage of image recognition [21]. CNN's are mainly composed of two parts: the convolutional and pooling layers. The convolutional layer extracts the image features from the filter matrix. In Figure 3, assume that matrix $5 \times 5$ has dimensions $(M_a, N_a)$ as $A(m,n)$ and filter matrix $3 \times 3$ has dimensions $(M_b, N_b)$ as $B(i-m, j-n)$. When the block calculates the full output size, the equation for the convolutional operation result $C(i,j)$ is:

$$C(i,j) = \sum_{m=0}^{(M_a-1)} \sum_{n=0}^{(N_a-1)} A(m,n) * B(i-m, j-n), \tag{1}$$

where $0 \leq i < M_a + M_b - 1$ and $0 \leq j < N_a + N_b - 1$.

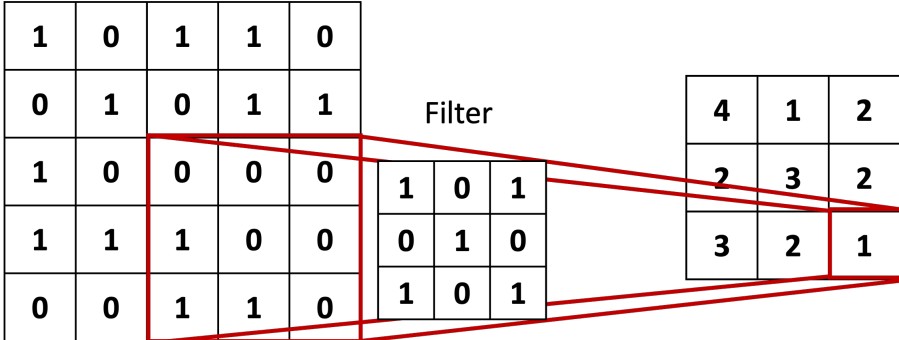

**Figure 3.** The convolutional operation and the numbers 0–4 are from Equation (1).

The pooling layer mainly reduces the dimension of the feature matrix, and the convolutional layer extracts it. It retains important features and removes noise, preventing overfitting. Pooling includes two methods: max pooling and means pooling. Figure 4 shows how the max-pooling operation works, taking the maximum value of the kernel matrix. CNN models can mimic human-vision systems by convolutional layer and pooling layer. The end of the model is a fully connected layer that compiles the feature matrix extracted by previous layers to form the final output.

The primary model of the financial vision is GAF-CNN [22]. The GAF-CNN is from GAF and CNN. The model structure is in Figure 5. The GAF-CNN model encodes time-series data using the Gramian Angular Field (GAF) method and then classifies the patterns using the convolutional neural network (CNN) model. The financial vision helps people interpret the data more directly. However, it is not considered in many traditional statistical or machine learning methods. In finance, a candlestick chart is a visualization method that can improve market analyses, such as seeking potential opportunities.

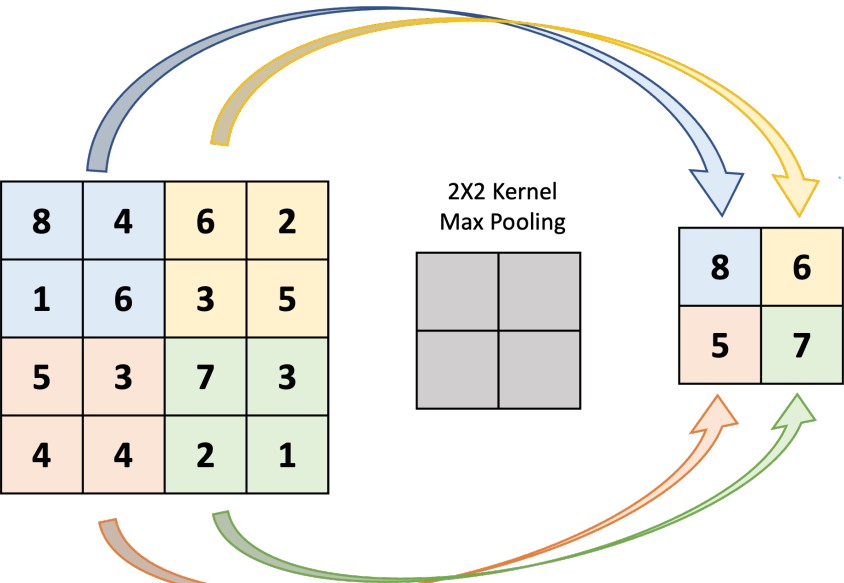

**Figure 4.** The max-pooling operation. There are four sub-matrices in the left matrix. The max-pooling operation (middle matrix) takes every maximum value of sub-matrices to the right matrix.

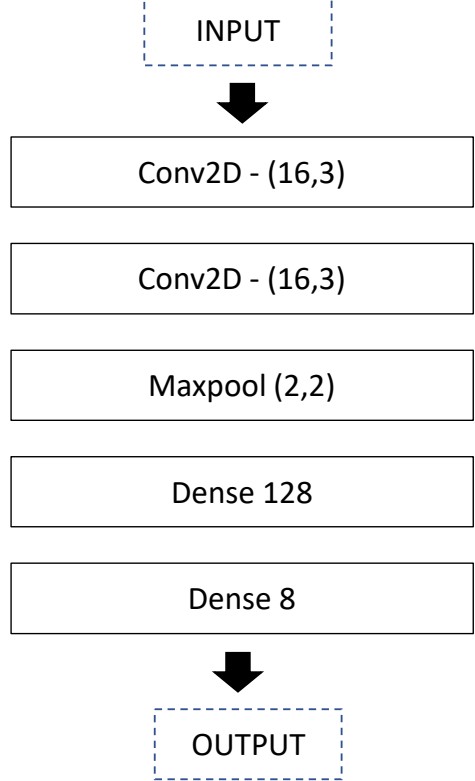

**Figure 5.** The construction of GAF-CNN model.

According to the financial vision applications, the technology can understand the critical components of a candle and what they indicate to apply candlestick chart analyses to a trading strategy [23]. The financial vision recognizes candlesticks automatically.

### 2.3. Differential Privacy

Dwork et al. proposed Differential Privacy (DP) to reduce the possibility of data leakage by adding noise [16]. It can prevent differential attacks, linkage attacks, and

reconstruction attacks. The model adds noise by using several mechanisms, such as the Laplace mechanism, Gaussian mechanism [16], or exponential mechanism [24]. The Laplace or Gaussian mechanism is suitable for numerical data, and the exponential tool is ideal for non-numeric data. Google applied the DP mechanism with deep learning in 2016, thereby reducing the possibility of the leakage of training data [25]. Many famous technology companies such as Apple [26] and Microsoft [27] are currently applying DP to protect data.

### 2.4. Model Attack

The application of deep learning models is becoming more widespread. Some attackers begin to decipher the training data by using the model's information, such as the parameters and gradients. There are two types of model attacks [28]. The first incudes reconstruction attacks to find out the attributes of the training data [29,30]. The second includes membership inference attacks. The attacker's goal is to find specific information in the training data [31–33]. The previous research published by M Fredrikson [12] is a type of membership inference attack. Moreover, membership inference attacks include white-box and black-box membership inference attacks. The model's entire structure needs a gradient in a white-box setting, and a black-box set can attack the model only through inputs and outputs. Combining black and white boxes to attack model makes the attack more effective [34,35]. With the development of attack techniques, it is more challenging to protect private data. Differential privacy is a method that can be used to protect data from being leaked due to model attacks in many fields.

## 3. Methods

### 3.1. Experimental Design

Although differential privacy can provide data privacy guarantees, it will also cause problems such as difficulty in data interpretation or a decrease in the model's accuracy. Thus, the noise level needs to be considered. Currently, the noise setting of differential privacy is inconclusive. This research aims to find a balance between privacy and accuracy. The four steps of the experimental procedure are in Figure 6. The first step is to build a GAF-CNN model as the baseline model. The second step is to build DP models with noise scales $0.1, 0.3, 0.5, 0.7, 1$. The settings are the same as the baseline model, except that the optimizer uses DP-SGD instead of SGD. The epsilons are calculated from different noise scales. The third step compares the accuracy of different noise scale models and baseline models. The last step is to use the white-box attack to attack the baseline and DP models. Then, the probability of data leakage in the training data is compared to understand the privacy guarantee in the baseline and DP models under different noise scales.

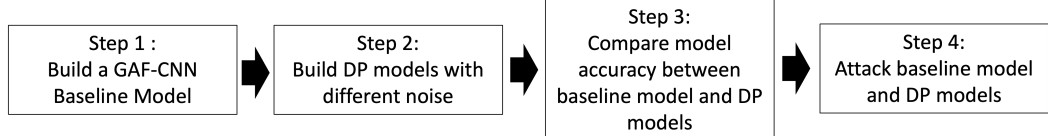

**Figure 6.** Experiment process.

### 3.2. Build a GAF-CNN Baseline Model

We chose GAF-CNN instead of the CNN model because the candlestick charts contain time-series data. It is hard to find essential features in CNN models. However, the GAF-CNN model found the parts easily. We take the morning star pattern as an example (Figure 7). There is a downtrend, and the trend's end is the pattern of two long candlesticks sandwiching a short one. Figure 8 shows the features found in the second conversion layer of the GAF-CNN model. In the CNN model of Figure 9, the downtrend is not here, and some outputs have no attributes. The GAF-CNN model found some particular patterns. Hence, we chose the GAF-CNN model. After introducing the GAF-CNN model, we will introduce some methods that can use the model to define the training data.

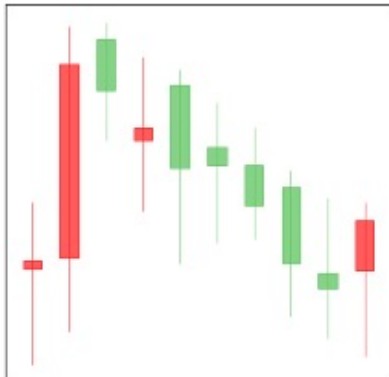

**Figure 7.** An example of morning star pattern. The red indicates the candlestick's close price is higher than the open price. The green shows the close price of these candlesticks is lower than the open price.

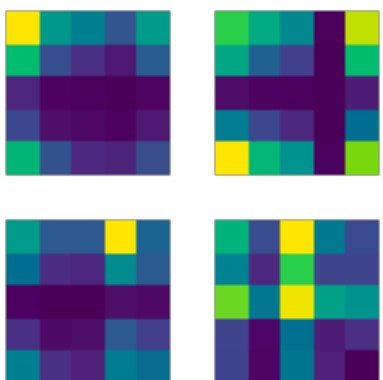

**Figure 8.** GAF-CNN output.

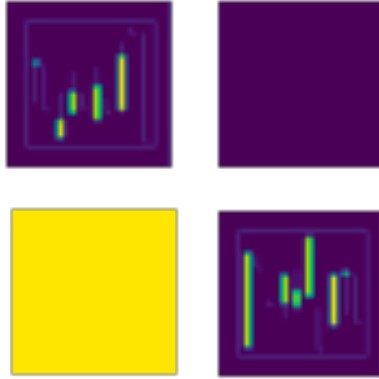

**Figure 9.** CNN output.

We elaborate a model using candlestick patterns and the time-series data as training data for the two phases. Firstly, the GASF method encodes time-series data. Secondly, a CNN model is trained using the encoded matrix. The CNN model has two convolutional layers and a fully connected layer [22]. We use the python package and optuna to search for the best parameters for 100 trains—the baseline model parameters in the Table 1. Moreover, we use the SGD optimizer with momentum.

**Table 1.** The best baseline model hyperparameters.

| Hyperparameter | Value |
|:---:|:---:|
| Learning rate | 0.0006 |
| Momentum | 0.9 |
| Batch size | 100 |
| Epoch | 100 |

There are three steps to encoding time-series data with GAF, shown in the Figure 10 [22]:

**Step 1:** Normalize time-series data between [0,1] using minimum–maximum scaling in Equation (2).

$$\widetilde{x}_i = \frac{x_i - \min(X)}{\max(X) - \min(X)} \tag{2}$$

**Step 2:** Represent the normalized time-series data with the polar coordinate system.

$$\phi = \arccos(\widetilde{x}_i), 0 \leq \widetilde{x}_i \leq 1, \widetilde{x}_i \in \widetilde{X}$$
$$r = \frac{t_i}{N}, t_i \in \mathbb{N} \tag{3}$$

**Step 3:** Adopting Equation (4), create a GASF matrix with the cosine of the summation of the angles.

$$\mathrm{GAF} = \cos(\phi_i + \phi_j)$$
$$= \begin{bmatrix} \cos(\phi_1 + \phi_1) & \cdots & \cos(\phi_1 + \phi_n) \\ \cos(\phi_2 + \phi_1) & \cdots & \cos(\phi_2 + \phi_n) \\ \vdots & \ddots & \vdots \\ \cos(\phi_n + \phi_1) & \cdots & \cos(\phi_1 + \phi_n) \end{bmatrix} \tag{4}$$

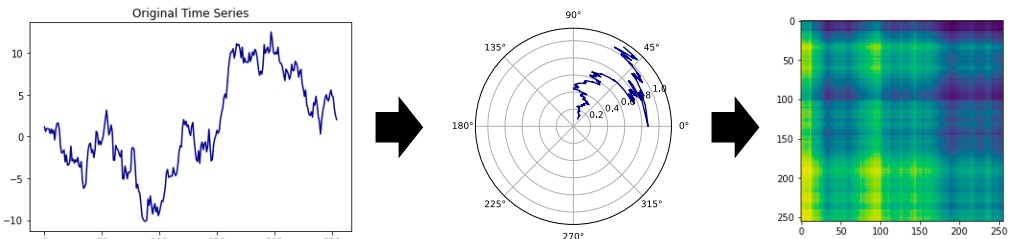

**Figure 10.** The GASF mechanism.

*3.3. Build DP Models with Different Noise*

The DP consists two parameters:

1. $\epsilon$: privacy loss;
2. $\delta$: probability of violating DP mechanism.

Let the notation be DP:$(\epsilon, \delta)$ in the Equation (5). An algorithm $M$ is with DP:$(\epsilon, \delta)$, where $\epsilon \geq 0$ and $0 \leq \delta \leq 1$.

A large $\delta$ extends the limitation of the total differential privacy [36]. $D$ and $D'$ differ in only one datum for any two neighboring data sets, and $S$ is an output of $M$ [16].

Privacy loss $\epsilon$ represents the probability of algorithm $M$ in obtaining the same output on neighboring data sets. The smaller the $\epsilon$, the more superior the privacy protection. When $\epsilon = 0$, the probability distribution of the output of algorithm $M$ for $D$ and $D'$ is the

same. The output of algorithm $M$ cannot reflect any useful information about the data set. $\epsilon$ also reflects the availability of data. The smaller $\epsilon$ is, the lower the availability of data.

$$Pr[M(D) \in S] \leq e^{\epsilon} \cdot Pr[M(D') \in S] + \delta \quad (5)$$

Due to the complexity of the DP:$(\epsilon, \delta)$ of $\epsilon$, a differential privacy variant Rényi Differential Privacy (RDP) later evolved [37]. The definition of RDP is in Equation (6). When $\alpha = \infty$, RDP:$(\alpha, \epsilon)$ and DP:$\epsilon$ are equal. DP can convert to RDP and vice versa. If algorithm M is satisfied with RDP, then, for all $\delta$, M is also satisfied with

$$DP : (\epsilon - \frac{\log \delta}{\alpha - 1}, \delta)$$

and

$$D_{\alpha}(M(D) \| M(D')) \leq \epsilon. \quad (6)$$

In this study, we use the Differentially Private Stochastic Gradient Descent (DP-SGD) proposed by Google, which applies differential privacy to gradients. In addition to Stochastic Gradient Descent (SGD), other first-order optimization methods are also applicable, such as AdaGrad or SVRG [25].

Google proposed the DP-SGD mechanism, adding noise to the gradient [25]. DP-SGD makes two modifications to the gradient. The first is to limit the gradient according to the L2 Norm to reduce the sensitivity of each training point. The second is to add random noise with the Gaussian mechanism to the gradient, making it more challenging to match the specific training data by comparing the gradient. The detailed process is described in Algorithm 1 [25].

---

**Algorithm 1** Differentially Private Stochastic Gradient Descent (DP-SGD).

---

**Load** training data $X = \{x_1, x_2, ..., x_N\}$
**Set** loss function $\mathcal{L}(\theta) = \frac{1}{N} \sum_{i=1}^{N} \mathcal{L}(\theta, x_i)$.
**Set** learning rate $\eta_t$, noise scale $\sigma$, group size $L$, and gradient norm bound $C$.
**Initialize** $\theta_0$ randomly.
**for** $t$ in $T$ **do**
　　Random sample a $L_t$ via sampling probability $\frac{L}{N}$.
　　**Step 1. Compute and clip gradient**
　　**for** $i$ in $L_t$ **do**
　　　　compute $g_t(x_i) \leftarrow \nabla_{\theta_t} \mathcal{L}(\theta_t, x_i)$
　　　　$\bar{g}_t(x_i) \leftarrow g_t(x_i) / max(1, \frac{\|g_t(x_i)\|_2}{C})$
　　**end for**
　　**Step 2. Add random noise**
　　$\tilde{g}_t \leftarrow \frac{1}{L}(\sum_{i=1}^{L} \bar{g}_t(x_i) + \mathcal{N}(0, \sigma^2 C^2 I))$
　　**Step 3. Gradient descent**
　　$\theta_{t+1} \leftarrow \theta_t - \eta_t \cdot \tilde{g}_t$
**end for**
**Step 4. Compute privacy cost**
**Output** $\theta_T$ and compute the privacy cost $(\varepsilon, \delta)$ with a privacy accounting method.

---

### 3.4. Compare Model Accuracy between Baseline Model and DP Models

According to Abadi, Martin et al. [25], we set the clip norm to 1 and 1.5, the noise to be between 1 and 0.1, and $\delta = 0.00001$. Then, the model's accuracy between different noises was compared and the privacy loss after model training was computed. The DP-SGD package is available from https://github.com/tensorflow/privacy/tree/master/tutorials, accessed on 1 June 2022.

### 3.5. Attack Baseline Model and DP Models

We evaluate the model's performance in the accuracy of the privacy model compared to the baseline model and compare the privacy between different noises. We also assess privacy in two ways: the privacy loss $\epsilon$ and the white-box attack result.

1.  Privacy loss ($\epsilon$):
    The privacy loss cannot be directly specified when setting parameters, and it has to be calculated by fixing the delta and noise size. Each gradient update will bring some privacy loss during the training process; thus, we must add up these privacy losses. The $\epsilon$ of DP-SGD makes calculations through the moments' account using Equation (7), which is shown below.

$$\sigma = \frac{\sqrt{2\log\frac{1.25}{\delta}}}{\epsilon} \tag{7}$$

2.  White-box attack result:
    Attack the baseline and DP-SGD models using the white-box attack model proposed by Jiayuan Ye et al. [34]. The code source is available from https://github.com/privacytrustlab/ml_privacy_meter, accessed on 1 June 2022. The attack model will show the probability of whether the input data are in training data or not. We will plot the probabilities by referring to Figure 11 from the ml_privacy_meter GitHub. It shows the training data probabilities and non-training data probabilities. The higher training data probabilities show that the model has predicted a higher likelihood and that the data are part of the training data. It can demonstrate the effectiveness of the DP-SGD model in protecting the training data from the attack result.

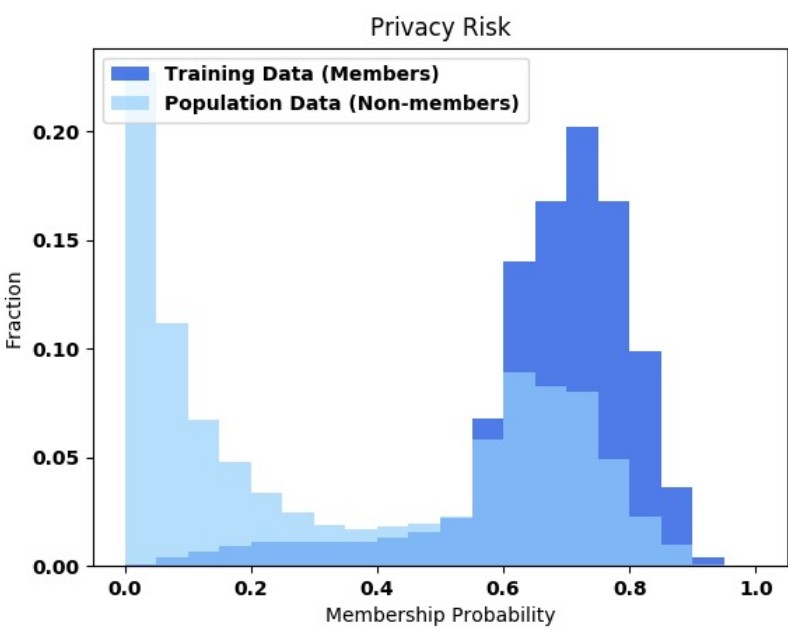

**Figure 11.** Example of privacy risk for training data.

## 4. Results

### 4.1. Data Illustration

The data and code are in https://github.com/pecu/FinancialVision, accessed on 1 June 2022. The project folder is "Financial Vision Based Differential Privacy Applications." We use EUR/USD price data to label eight candlestick patterns from 1 January 2010 to 1 January 2018. The training set and testing set ratio are 3:1—a total of 12,000 data of 1500 per label in the training set. In the validation set, there is a total of 1600 data of 200 per label. The testing set has a capacity of 4000 data of 500 per label. Moreover, convert the time-series

information into the GASF. The eight patterns include Evening Star, Morning Star, Bearish Engulfing, Bullish Engulfing, Shooting Star, Inverted Hammer, Bearish Harami, and Bullish Harami patterns. Morning Star, Bullish Engulfing, Inverted Hammer, and Bullish Harami are bullish patterns, and the other four are bearish patterns. The introduction of the eight classes below refers to the major candlesticks signals [38]. In the candlestick pattern examples, the red candlestick represents bullish, and the green candlestick represents bearish, which is the same as the Taiwan stock market. Moreover, the left side is the time series of opening, high, low, and closing prices (OHLC) converted to GASF. The right side is the time series of the closing prices, upper shadow, lower shadow, and real-body (CULR) converted into GASF.

1.  Evening Star (Figure 12) consists of three candlesticks, the first is a long red candlestick, the second is a very short candlestick, and the third is a long green candlestick. This pattern is bearish.

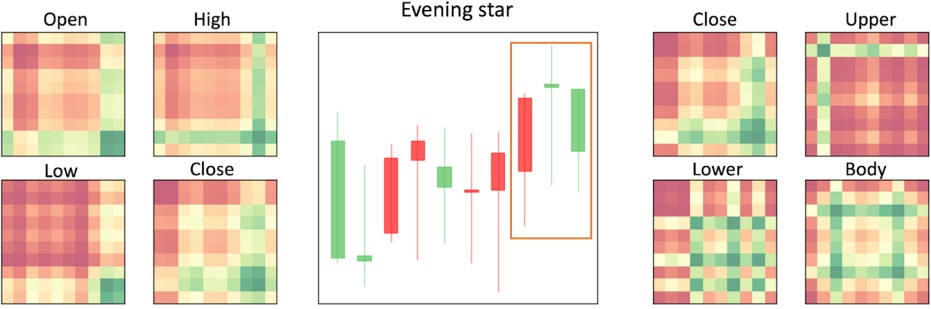

**Figure 12.** Example of Evening Star pattern and GASFs. The middlemost figure is the original candlesticks. The red and green candlesticks represent closing prices are higher than the opening prices are not. The left and right sides are the heatmaps of GASFs. The range of the heatmap is $(-1, 1)$, where red equals $-1$ and green equals $1$.

2.  Morning Star (Figure 13) is composed of three candlesticks. Opposite of Evening Star, the Morning Star represents a bullish pattern. The first candlestick is a long green one. The second is very short, and the third is a long red. The longer the third one is, the greater the upward trend.

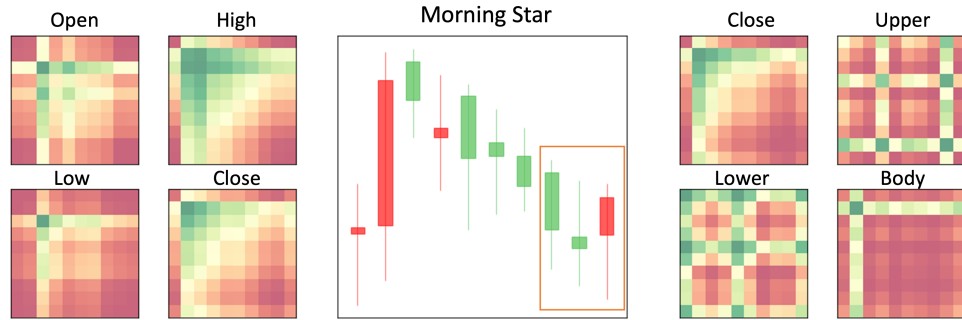

**Figure 13.** Example of Morning Star pattern and GASFs. The middlemost figure is the original candlesticks. The red and green candlesticks represent closing prices are higher than the opening prices are not. The left and right sides are the heatmaps of GASFs. The range of the heatmap is $(-1, 1)$, where red equals $-1$ and green equals $1$.

3.  Bearish Engulfing (Figure 14) consists of two candlesticks, usually after an uptrend. The first bar is red, the second bar is green, and the first body will be smaller than the second. Bearish Engulfing represents a bearish pattern.

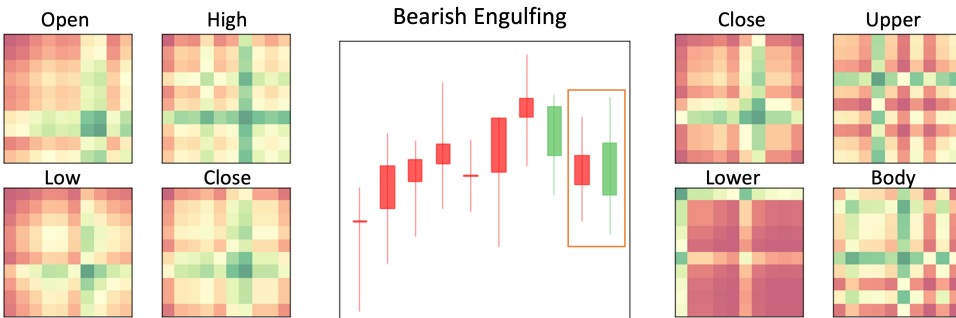

**Figure 14.** Example of Bearish Engulfing pattern and GASFs. The middlemost figure is the original candlesticks. The red and green candlesticks represent closing prices are higher than the opening prices are not. The left and right sides are the heatmaps of GASFs. The range of the heatmap is $(-1, 1)$, where red equals $-1$ and green equals 1.

4.   Bullish Engulfing (Figure 15) is the opposite of Bearish Engulfing, indicating a bullish pattern. It consists of two candlesticks; the first green candlestick body is smaller than the second red candlestick.

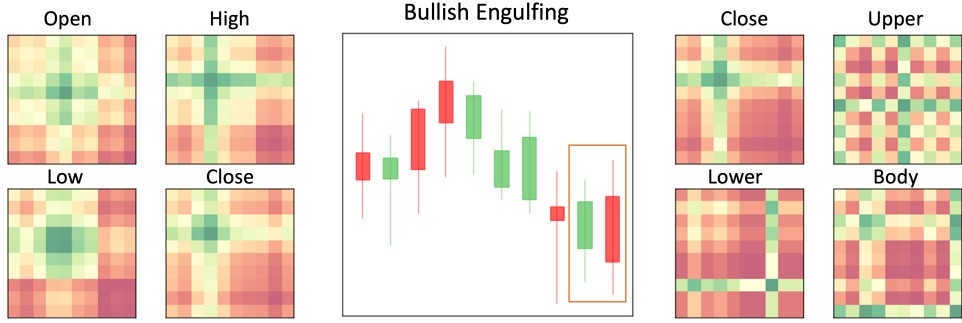

**Figure 15.** Example of Bullish Engulfing pattern and GASFs. The middlemost figure is the original candlesticks. The red and green candlesticks represent closing prices are higher than the opening prices are not. The left and right sides are the heatmaps of GASFs. The range of the heatmap is $(-1, 1)$, where red equals $-1$ and green equals 1.

5.   Shooting Star (Figure 16) consists of two candlesticks, which usually appear behind the uptrend. The first candlestick is red, and the color of the second one is not essential, as long as it is short enough and has a long upper shadow. This pattern indicates a bearish pattern.

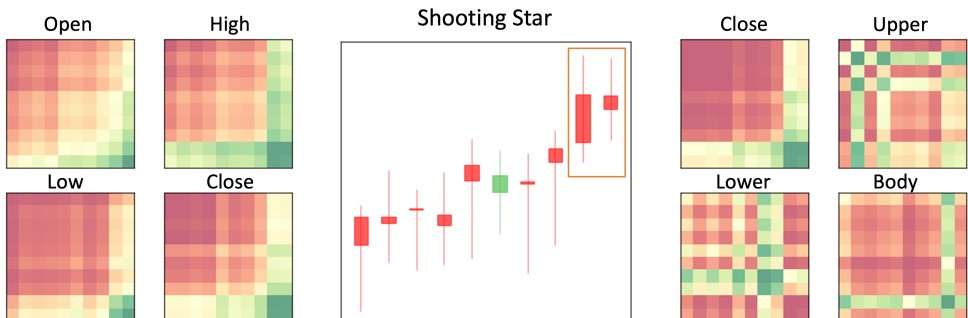

**Figure 16.** Example of Shooting Star pattern and GASFs. The middlemost figure is the original candlesticks. The red and green candlesticks represent closing prices are higher than the opening prices are not. The left and right sides are the heatmaps of GASFs. The range of the heatmap is $(-1, 1)$, where red equals $-1$ and green equals 1.

6.      Inverted Hammer (Figure 17) is the opposite of a shooting star, which represents a bullish pattern. It usually appears behind the downtrend. The first candlestick is green, and the color of the second candlestick is not essential. However, the second candlestick should be short enough and the upper shadow should be long enough.

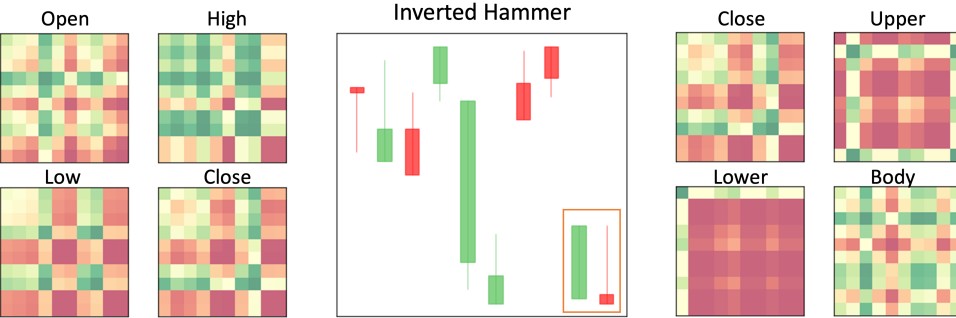

**Figure 17.** Example of Inverted Hammer pattern and GASFs. The middlemost figure is the original candlesticks. The red and green candlesticks represent closing prices are higher than the opening prices are not. The left and right sides are the heatmaps of GASFs. The range of the heatmap is $(-1, 1)$, where red equals $-1$ and green equals 1.

7.      Bearish Harami (Figure 18) is composed of two candlesticks, with the first candlestick being a longer one and the second candlestick being a shorter green one. The body of the second candlestick must be more concise than the first candlestick. This pattern indicates a signal of price reversal from an uptrend to a downtrend.

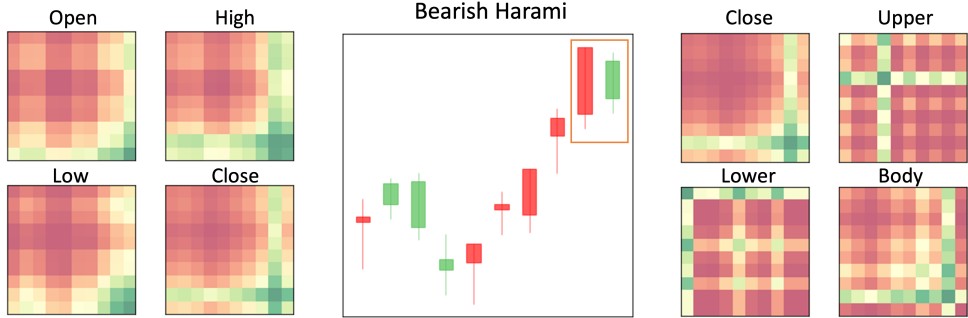

**Figure 18.** Example of Bearish Harami pattern and GASFs. The middlemost figure is the original candlesticks. The red and green candlesticks represent closing prices are higher than the opening prices are not. The left and right sides are the heatmaps of GASFs. The range of the heatmap is $(-1, 1)$, where red equals $-1$ and green equals 1.

8.      Bullish Harami (Figure 19) is the opposite of Bearish Harami. It is a sign of price reversal from a downtrend to an uptrend. Bullish Harami consists of a first longer green candlestick and a second shorter red candlestick. The second one is engulfed by the first one.

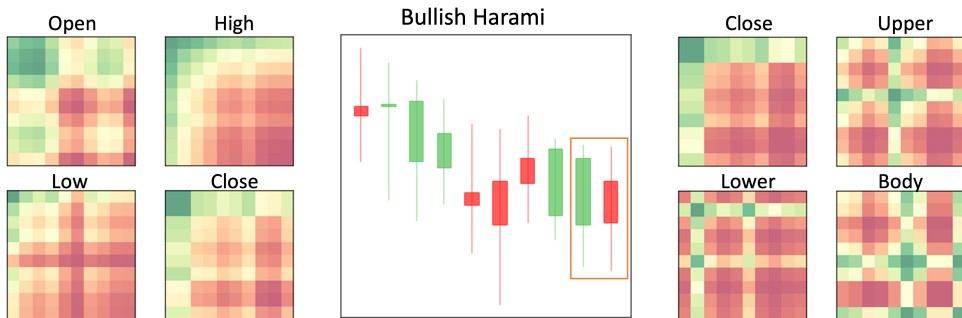

**Figure 19.** Example of Bullish Harami pattern and GASFs. The middlemost figure is the original candlesticks. The red and green candlesticks represent closing prices are higher than the opening prices are not. The left and right sides are the heatmaps of GASFs. The range of the heatmap is $(-1, 1)$, where red equals $-1$ and green equals 1.

*4.2. DP Result*

The test accuracy of the Baseline model is 96.13%, and the training process is shown in Figure 20.

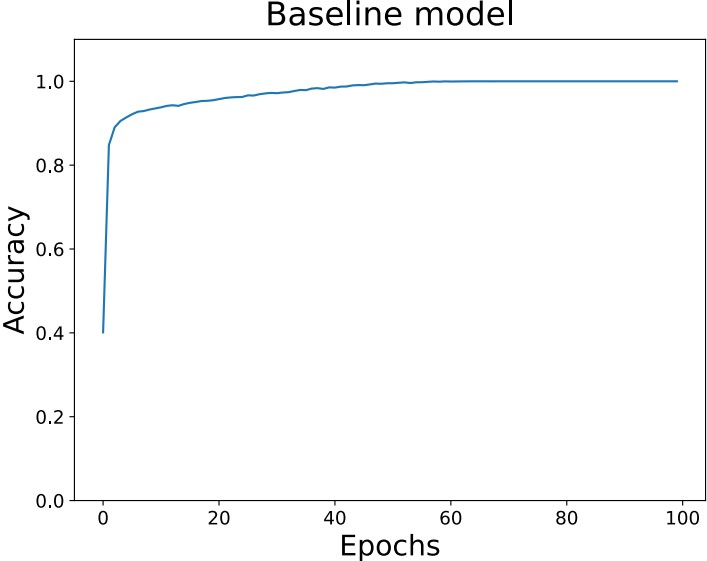

**Figure 20.** Baseline model training process.

We will compare the test accuracies between the baseline model and the DP models with different noises. We will also compare the performance of the DP model with different gradient clippings. The training process of DP Models with different gradient clipping is in Figure 21. The result shows that accuracy is more stable when the gradient clipping = 1.5 than a gradient clipping of 1. The model's accuracy with different noises does not differ much during the training process in the same clipping. The $\epsilon$ and test accuracy for each DP Model is in Table 2.

We defined the probability of training data from being defined correctly by the attack model as the recognition rate. The recognition rate of the baseline model is 66.89%. The recognition rates of DP models are in Table 2. The recognition rate decreases as the noise increases. The noise increased from 0.1 to 1. The recognition rate decreased from 62.11% to 46.04%.

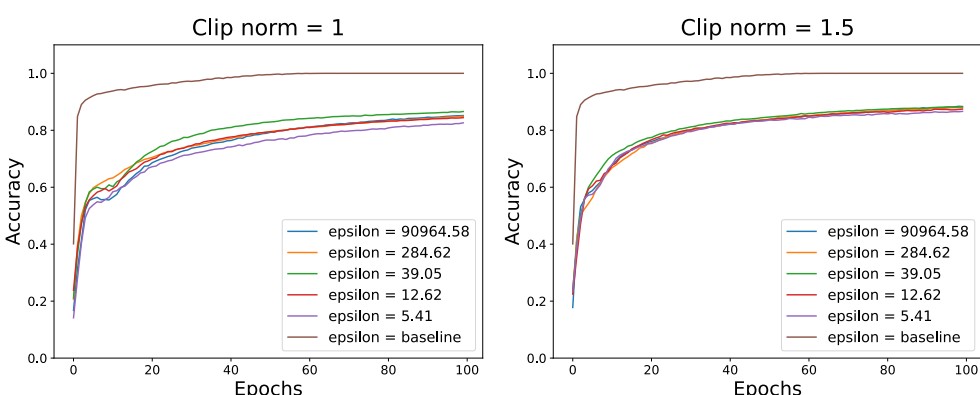

**Figure 21.** DP model training process.

**Table 2.** Test accuracies in different gradient clipping and noise.

| Clipping Noise ($\epsilon$) | 1 | 1.5 | Recognition Rate |
|---|---|---|---|
| 0.1 (90,964.58) | 88.95% | 91.15% | 62.11% |
| 0.3 (284.62) | 87.48% | 91.25% | 58.21% |
| 0.5 (39.05) | 87.70% | 90.75% | 51.40% |
| 0.7 (12.62) | 87.15% | 90.45% | 49.65% |
| 1 (5.41) | 85.98% | 89.23% | 46.04% |

*4.3. Attack Results*

We use the ml_privacy_meter package to attack the baseline and the DP models separately to understand the probability distribution of recognized training data between these models. Figure 22 shows the absence of DP Mechanism. It has a high probability of correctly identifying whether it is training data. In the DP models, the probability of training data being recognized decreases when the noise increases. The larger the noise, the less chance there is to distinguish between training and non-training data. When noise is 0.1, the probability of correctly identifying training data is near 1. The probability that the data of the non-training data are defined as the training data is near 0 (Figure 23), similarly to the the result of the baseline model. Figure 24 shows that when noise = 0.3, the probability of correctly identifying training data shifts to the left slightly, and the probability of the non-training data is defined as the training data shifting to the right slightly, showing that when the noise proceeds from 0.1 to 0.3, the accuracy of labeling training data and non-training data becomes a little lower. Figure 25 shows when noise = 0.5, the percentage of the correct identification of training data becomes lower, and the probability is also reduced. Non-training data also have a greater probability of being misjudged as training data. Noise = 0.7 and noise = 1 are shown in Figures 26 and 27, and the probability distributions of training data and non-training data are very similar; that is, it is difficult to identify the training data correctly.

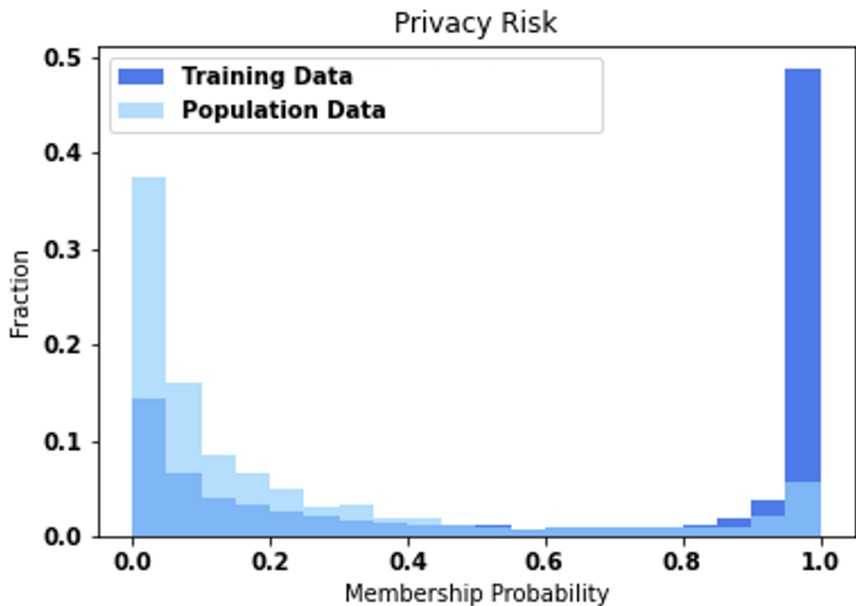

**Figure 22.** Privacy risk of the baseline model.

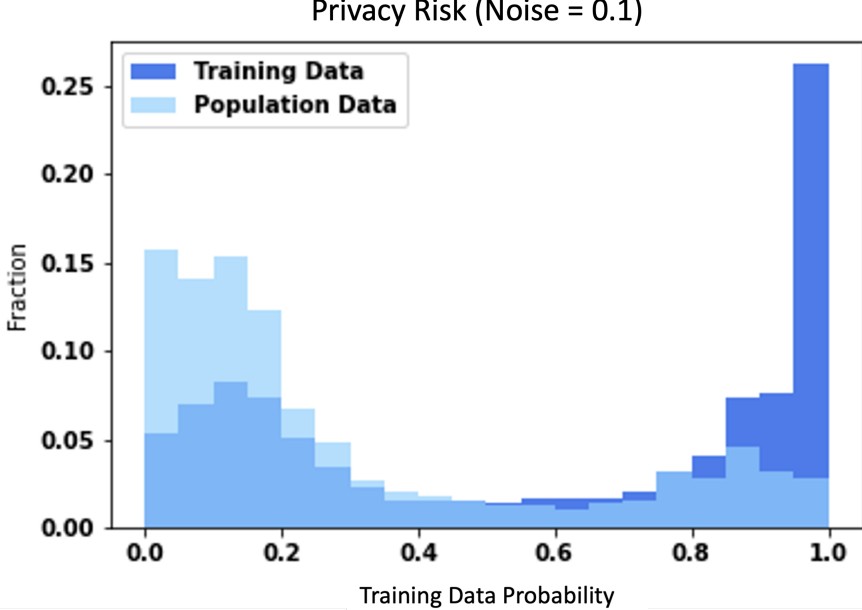

**Figure 23.** Noise = 0.1.

We also convert the recognition rate distribution into a violin plot. Figure 28 shows that only a tiny proportion is distinguished in the baseline model, and most will be correctly identified. As noise becomes more significant, the probability of correctly identifying training data decreases gradually, as shown in Figure 29. When noise = 1, more training data are not correctly specified. The recognition rate in the baseline model is 66.89%. When noise is 1, the recognition rate is only 46.04%, which a drop of nearly 21%. The attack results show that a differential privacy mechanism can protect the training data effectively. The higher the privacy loss ($\epsilon$), the higher the probability of the training data being the same as Dwork and Roth's claim [39]. However, the difference is that if $\epsilon$ is not too high, there is still a privacy guarantee even if it is higher than 1.

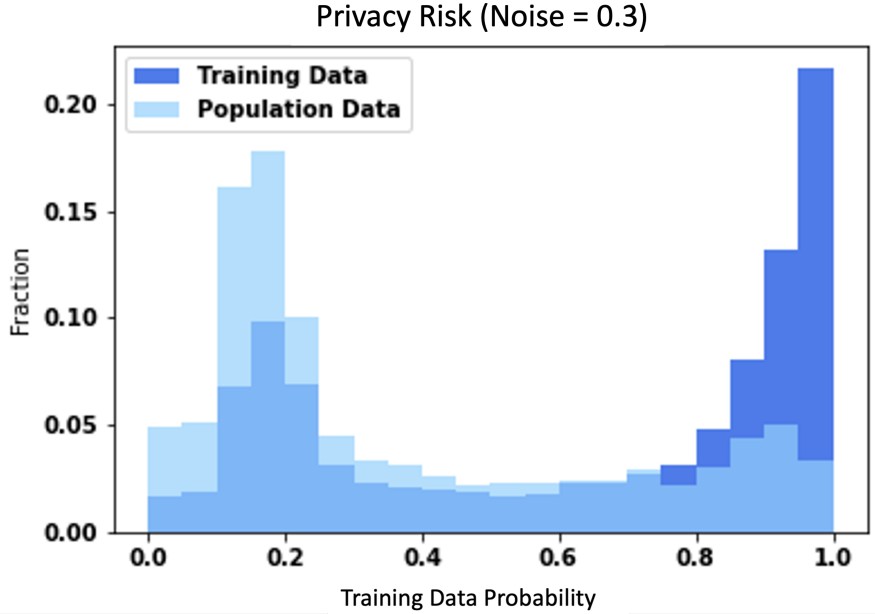

**Figure 24.** Noise = 0.3.

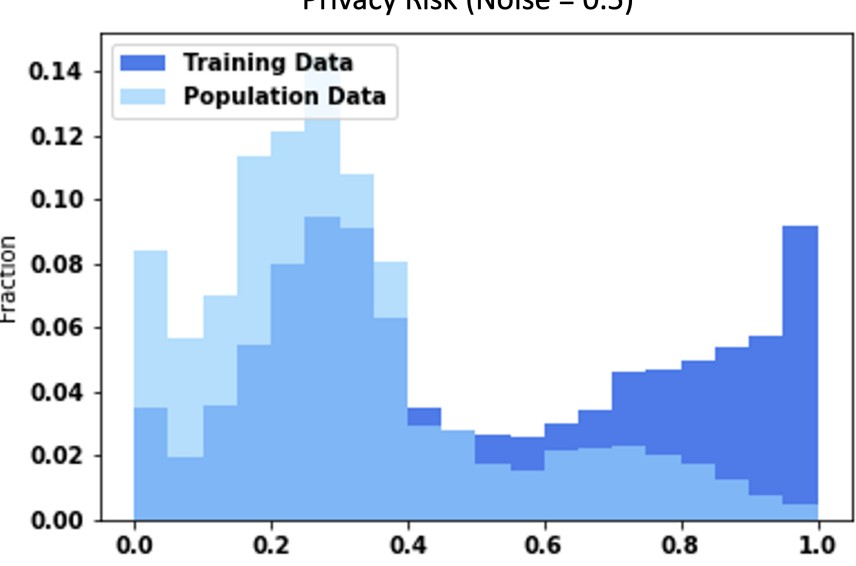

**Figure 25.** Noise = 0.5.

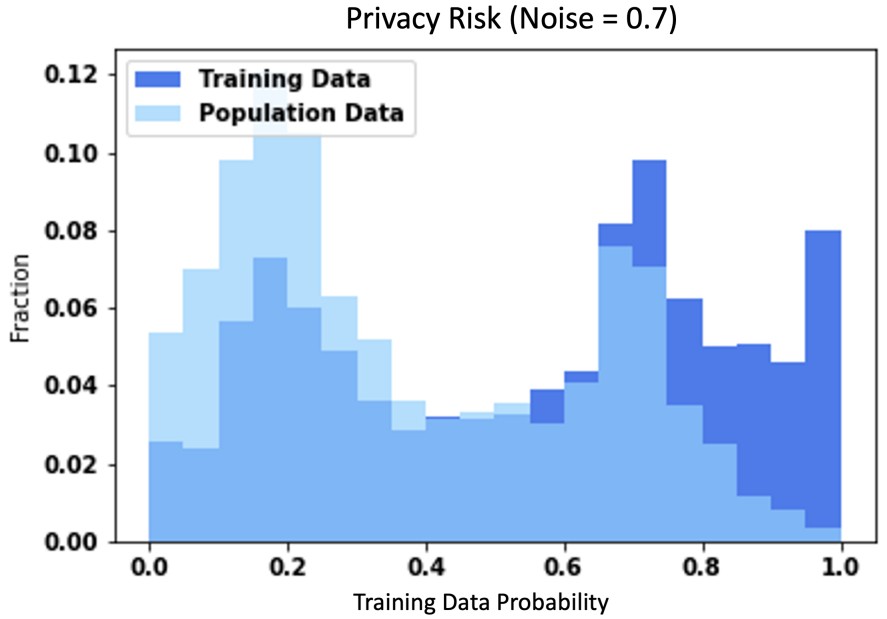

**Figure 26.** Noise = 0.7.

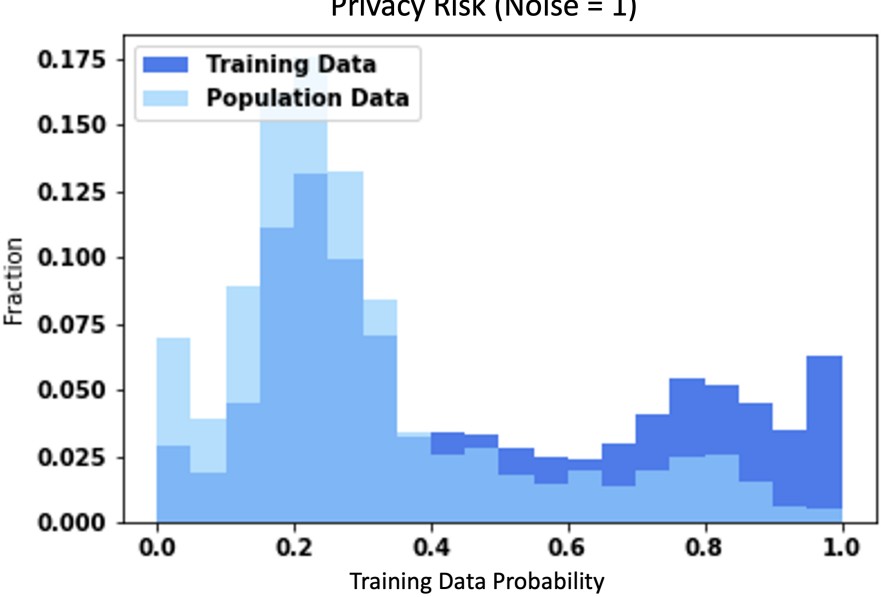

**Figure 27.** Noise = 1.

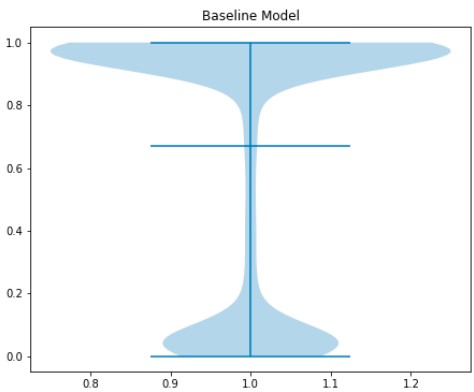

**Figure 28.** Recognition rate of the baseline model.

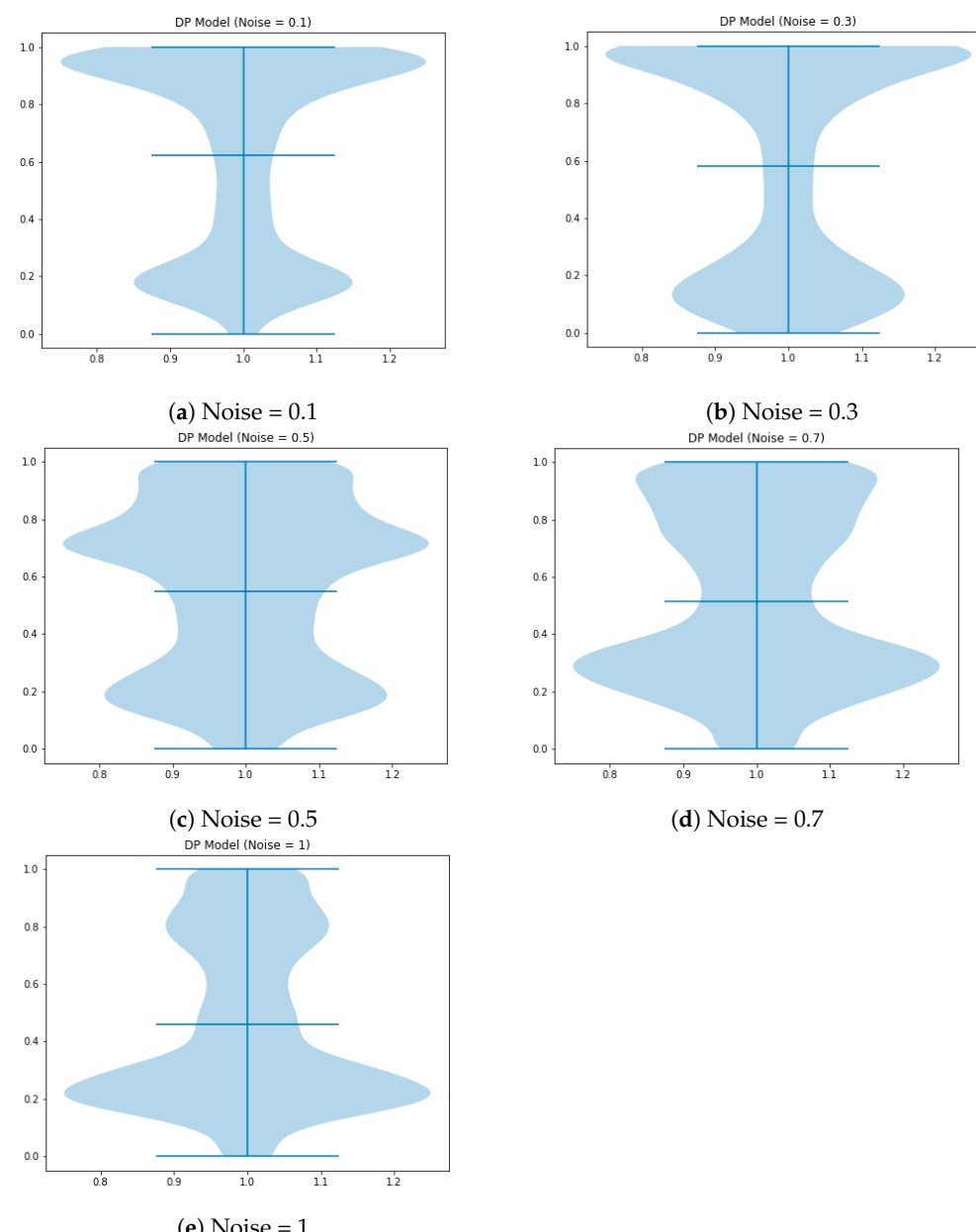

(**a**) Noise = 0.1

(**b**) Noise = 0.3

(**c**) Noise = 0.5

(**d**) Noise = 0.7

(**e**) Noise = 1

**Figure 29.** Recognition rate of DP models.

## 5. Discussion

When noise size = 0.1 and 0.3, $\epsilon$ = 90,964.58 and 284.62, respectively. The value of $\epsilon$ is so high that there is almost no privacy guarantee. Therefore, we will not discuss the two models. When noise is 0.5, $\epsilon$ = 39.5, test accuracy = 90.75%, which is about 5% lower than baseline model's test accuracy. When noise is 0.7, $\epsilon$ = 12.62, test accuracy = 90.45% , which is about 6% lower than the baseline model. When noise is 1, $\epsilon$ = 5.41, test accuracy = 89.23%, and it is about 7% lower than the baseline model. We can find that even when noise is 1, the accuracy does not decrease too much.

Financial vision contains sensitive information flagged by experts and could cause significant damage if leaked. Fortunately, differential privacy prevents data leakage. However, the differential privacy mechanism protects data by adding noise to the model, inevitably leading to model accuracy degradation or data distortion problems. The balancing data's readability does not provide a conclusion when we train the differential privacy model. Dwork and Roth recommend $\epsilon \leq 1$ to obtain a privacy guarantee [39]. Nevertheless, using such a small $\epsilon$ will cause difficulties in data interpretation. A higher $\epsilon$ is usually selected to balance privacy guarantee and data readability. For example, the $\epsilon$ standard used by Google and Apple is between 6 and 14 [17,40]. Similar results are in our experiments. Model accuracy decreases by approximately 6% with $\epsilon$ = 12.62 and only 7% with $\epsilon$ = 5.41. The accuracy reduction is tolerable, and the privacy guarantee of the training data is greatly improved.

On the other hand, there are excessively different results of differential privacy for different data types [25]. The details are in Table 3. In MNIST, compared to the baseline model, when $\epsilon$ = 2, the accuracy only decreases around 3.3%. However, in CIFAR-10, the accuracy dived from 96.5 to 70% when $\epsilon$ = 4. In our data, the accuracy of the model of $\epsilon$ = 5 is 7% lower than the baseline model. Although not as good as MNIST, it is still acceptable.

**Table 3.** The accuracy between different data.

| MNIST | | CIFAR-10 | | Our Data | |
|---|---|---|---|---|---|
| $\epsilon$ | **Accuracy** | $\epsilon$ | **Accuracy** | $\epsilon$ | **Accuracy** |
| Baseline | 98.3% | Baseline | 96.5% | Baseline | 96.13% |
| 2 | 95% | 4 | 70% | 5.41 | 89.23% |
| 8 | 97% | 8 | 73% | 12.62 | 90.45% |

## 6. Conclusions

This is the first paper to balance privacy guarantees and data readability for financial visual data. We compare the results with different privacy for different data types in the Table 3. This comparison shows data on how to find a balance between privacy and accuracy with DP-SGD.

There are three contributions to this research. The first was to provide a variety of demonstrations of financial vision with a differential privacy process, which can be a reference for those who need it. The second provides parameter selection criteria. The third specifies different privacy losses ($\epsilon$) (that is, how much actual protections different $\epsilon$ can provide).

In the future, we can explore the effect of combining differential privacy mechanisms and federated learning in the financial field, such as the impact of updating the federated model by using multiple local model training with differential privacy. The differential privacy mechanism extends the entire FinTech in the future. Various financial institutions or even non-financial institutions can cooperate with privacy guarantees, improve the current limitations caused by data silos or cross-border cooperation, and maximize the value of data and benefits for both financial institutions and customers.

**Author Contributions:** Conceptualization, Y.-C.T.; Formal analysis, Y.-R.W.; Methodology, Y.-C.T.; Project administration, Y.-C.T.; Resources, Y.-C.T.; Software, Y.-R.W.; Supervision, Y.-C.T.; Visualization, Y.-R.W.; Writing—original draft, Y.-R.W.; Writing—review & editing, Y.-C.T. All authors have read and agreed to the published version of the manuscript.

**Funding:** This research received no external funding.

**Institutional Review Board Statement:** Not applicable.

**Informed Consent Statement:** Not applicable.

**Data Availability Statement:** The model is available on Github: https://github.com/pecu/FinancialVision. The folder of the repository is (The_Protection_of_Data_Sharing_for_Privacy_in_Financial_Vision). The used data is in the https://drive.google.com/file/d/1cCym8Re1aPDep29_cj9kUavCrYzpGV-U/view.

**Acknowledgments:** The authors are appreciated and grateful to the Jun-Hao Chen, Samuel Yen-Chi Chen, and PecuLab, Department of Technology Application and Human Resource Development, NTHU, for the enlightenment and support of the research.

**Conflicts of Interest:** The authors declare no conflict of interest.

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
