# Peer review of "The Protection of Data Sharing for Privacy in Financial Vision"

_applsci, doi:10.3390/app12157408_

Round 1
Reviewer 1 Report
This paper compares data security with the non-private and differential privacy financial visual models. In addition, it finds a balance between privacy protection with model accuracy. Thought the results show that when the privacy loss parameter ϵ is between 12.62 and 13 5.41, the privacy models can protect training data, and the accuracy does not drop too much.
To improve this work, the authors need to address the following issues:
1. The writing skills for this paper are quite poor. The paper needs a thorough proofreading and some of the sentences need to be rephrased.
2. The motivation for this paper and the contribution are not well explained or are not clear.
3. The discussion of relevant related is not clear.
The literature need to provide clearly the contribution of other author if possible it is good to have a comparison table on keywords.
Author Response
Point 1: The writing skills for this paper are quite poor. The paper needs a thorough proofreading and some of the sentences need to be rephrased.
Response 1: We restructured the paper following sections: Introduction, Materials and Methods, Results, Discussion, and Conclusions. We moved the experimental design part to the Methods and explained the analysis and the results in the Results. We also restructure the introduction. The main story includes three parts as followings:
-
Financial institutions are gradually moving towards a cooperative model of information sharing. The Financial Supervisory Commission plans to provide financial API services. After obtaining customer authorization, banks can query customer or transaction information through financial APIs
-
As the sharing of data increasing, privacy protection is a crucial issue. Many studies have considered the balance between big data and data privacy doing research.
-
This study compares data security with non-private and personal models of financial vision. And find a balance between privacy protection and model accuracy. Thought the results show that when the privacy loss parameter ϵ is between 12.62 and 13 5.41, the privacy models can protect training data, and the accuracy does not drop too much.
Point 2: The motivation for this paper and the contribution are not well explained or are not clear.
Response 2: The primary motivation is to address difficulties in data interpretation or a reduction in model accuracy. Although differential privacy can provide data privacy guarantees, it also brings problems. So the noise setting needs to be considered, and the noise setting for differential privacy is currently inconclusive. The main contribution is to find a balance between privacy and accuracy.
Point 3: The discussion of relevant related is not clear. The literature need to provide clearly the contribution of other author if possible it is good to have a comparison table on keywords.
Response 3: This is the first paper to balance privacy guarantees and data readability for financial visual data. We just need to compare the results with different privacy for different data types in Table 3. This comparison shows data on how to find a balance between privacy and accuracy with DP-SGD.

Reviewer 2 Report
The introduction is a little confusing. Some sentences appear almost standalone and the introduction it does not really flow. Rather, it seems to change direction in what it is trying to convey without providing alignment with the previous sentence or paragraph. This could be improved with a bit more weaving together of the concepts so that the reader can understand how these relate and understand to help capture the context and tell the 'story' would improve this.
The format of the paper could be improved by aligning this with the guidance provided for authors for this journal which suggests the following sections should be included: Introduction, Materials and Methods, Results, Discussion, Conclusions (optional). Here we see a 'Preliminary' section that explains some data processing methods which, presumably could account for a background review mixed with a bit of method. While there is a methodology section this is brief and really, the method is mixed in with the preliminary and the method so it is difficult to distinguish what is previous research that will be applied, how this will be applied and what is just explanation., This could be improved by starting off with explaining the theory before then outlining how this will be used in this work. This means that the experimental design should form part of the methodology before then moving on to explain what was done, the analysis and the results.
In Section 2.1, there is discussion about a candlestick chart that mentions colours and how these are used yet, the figure that accompanies this is in black and white. Similarly, there is mention of chapters, this is not a book but an article.
Overall, some interesting work but this needs to be reformatted and presented with better context and flow to fully make sense.
Author Response
Point 1: The introduction is a little confusing. Some sentences appear almost standalone and the introduction it does not really flow. Rather, it seems to change direction in what it is trying to convey without providing alignment with the previous sentence or paragraph. This could be improved with a bit more weaving together of the concepts so that the reader can understand how these relate and understand to help capture the context and tell the 'story' would improve this.
Response 1: We restructure the introduction. The main story includes three parts as followings:
-
Financial institutions are gradually moving towards a cooperative model of information sharing. The Financial Supervisory Commission plans to provide financial API services. After obtaining customer authorization, banks can query customer or transaction information through financial APIs.
-
As the sharing of data increasing, privacy protection is a crucial issue. Many studies have considered the balance between big data and data privacy doing research.
-
This study compares data security with non-private and personal models of financial vision. And find a balance between privacy protection and model accuracy. Thought the results show that when the privacy loss parameter ϵ is between 12.62 and 13 5.41, the privacy models can protect training data, and the accuracy does not drop too much.
Point 2: The format of the paper could be improved by aligning this with the guidance provided for authors for this journal which suggests the following sections should be included: Introduction, Materials and Methods, Results, Discussion, Conclusions (optional). Here we see a 'Preliminary' section that explains some data processing methods which, presumably could account for a background review mixed with a bit of method. While there is a methodology section this is brief and really, the method is mixed in with the preliminary and the method so it is difficult to distinguish what is previous research that will be applied, how this will be applied and what is just explanation., This could be improved by starting off with explaining the theory before then outlining how this will be used in this work. This means that the experimental design should form part of the methodology before then moving on to explain what was done, the analysis and the results.
Response 2: We restructured the paper following sections: Introduction, Materials and Methods, Results, Discussion, and Conclusions. We moved the experimental design part to the Methods and explained the analysis and the results in the Results.
Point 3: In Section 2.1, there is discussion about a candlestick chart that mentions colors and how these are used yet, the figure that accompanies this is in black and white. Similarly, there is mention of chapters, this is not a book but an article.
Response 3: In Section 2.1, the candlestick chart colors have two types. When the figure accompanying this is black and white, black means red, and white means green. We replace the chapters with sections.

Round 2
Reviewer 1 Report
The comments were addressed accordingly
Author Response
Thanks for Reviewer 1's Comments and Suggestions (The comments were addressed accordingly).
Reviewer 2 Report
This has been improved. There are a few areas that still need some work before this makes sense:
Line 30 - 94 what?
Line 36 … a lot of personal privacy.. Do you mean personal data?
Section 2 line 97-99 - This is a bit confusing. Am I correct in reading this to say that what you saying is that in Taiwan the interpretation of the candlestick is the opposite of the rest of the world? So that, in Taiwan red denotes decrease and green and increase whereas, in other jurisdictions, green denotes a decrease while red denotes an increase?
Line 158 - .. the noise level needs to be considered
Line 321 - does not make sense ..fortunately, differential privacy [something/what?] against data leakage
The results are interesting, would like to see a little more discussion beneath each section as to interpretation of each result and what that means in practice
Author Response
Response to Reviewer 2 Comments
Point 1: Line 30 - 94 what? Line 36 … a lot of personal privacy.. Do you mean personal data?
Response 1: We changed “personal privacy” with “personal data.”
Point 2: Section 2 line 97-99 - This is a bit confusing. Am I correct in reading this to say that what you saying is that in Taiwan the interpretation of the candlestick is the opposite of the rest of the world? So that, in Taiwan red denotes decrease and green and increase whereas, in other jurisdictions, green denotes a decrease while red denotes an increase?
Response 2: Yes, you are correct. The interpretation of the candlestick is the opposite of the rest of the world in Taiwan.
Point 3: Line 158 - .. the noise level needs to be considered
Response 3: We changed “So the noise setting needs to consider.” with “So the noise level needs to be considered.”
Point 4: Line 321 - does not make sense ..fortunately, differential privacy [something/what?] against data leakage
Response 4: The revised version is “Financial Vision contains sensitive information flagged by experts and could cause significant damage if leaked. Fortunately, differential privacy prevents data leakage.”
